# Importance of Magnesium Status in COVID-19

**DOI:** 10.3390/biology12050735

**Published:** 2023-05-18

**Authors:** Fernando Guerrero-Romero, Oliver Micke, Luis E. Simental-Mendía, Martha Rodríguez-Morán, Juergen Vormann, Stefano Iotti, Nikolina Banjanin, Andrea Rosanoff, Shadi Baniasadi, Guitti Pourdowlat, Mihai Nechifor

**Affiliations:** 1Biomedical Research Unit, Mexican Social Security Institute, Durango 34067, Mexico; guerrero.romero@gmail.com (F.G.-R.); luis_simental81@hotmail.com (L.E.S.-M.); rodriguez.moran.martha@gmail.com (M.R.-M.); 2Department of Radiation Therapy and Radiation Oncology, Franziskus Hospital, 33615 Bielefeld, Germany; strahlenklinik@web.de; 3Institute for Prevention and Nutrition, 85737 Ismaning, Germany; vormann@ipev.de; 4Department of Pharmacy and Biotechnology, Universita di Bologna, 40126 Bologna, Italy; stefano.iotti@unibo.it; 5National Institute of Biostructures and Biosystems, 00136 Rome, Italy; 6Institute of Hygiene and Medical Ecology, Faculty of Medicine, University of Belgrade, 11000 Belgrade, Serbia; nikolina.banjanin@med.bg.ac.rs; 7CMER Center for Magnesium Education & Research, Pahoa, HI 96778, USA; 8Tracheal Diseases Research Center, National Research Institute of Tuberculosis and Lung Diseases, Shahid Beheshti University of Medical Sciences, Tehran 198396-3113, Iran; sbaniasadi@yahoo.com; 9Chronic Respiratory Diseases Research Centre, National Research Institute of Tuberculosis and Lung Diseases, Shahid Beheshti University of Medical Sciences, Tehran 198396-3113, Iran; pourdowlat_g@yahoo.com; 10Department of Pharmacology, Gr. T Popa University of Medicine and Pharmacy, 700115 Iasi, Romania; mihainechif@yahoo.com

**Keywords:** COVID-19, SARS-CoV-2, magnesium, Mg, zinc, Zn, hypomagnesemia, dietary magnesium, magnesium deficit, serum magnesium, neurological/psychiatric aspects of COVID-19, inhaled magnesium therapy, pulmonary aspects of COVID-19

## Abstract

**Simple Summary:**

Magnesium is an essential nutrient, also called an essential mineral or element. Magnesium is vastly important in all life, with vital roles for the healthy functioning of the human immune, metabolic, neurological, psychological, and heart and circulatory systems. Our bodies cannot “make” magnesium; rather, we *must* get it from outside the body—from our environment, usually through food and water. In most modern cultures, magnesium intake has been low for decades. When the COVID-19 pandemic began, a group of international magnesium researchers were profoundly struck by the strong similarities between low magnesium status and the many risk factors for COVID-19, including older age, obesity, diabetes, kidney disease, cardiovascular disease, hypertension, and ailments that affect breathing. They began Zoom meetings to share ideas, data, and study plans and formed the MaGNet Global Magnesium Project. This review, written by several MaGNet members, discusses several peer-reviewed studies demonstrating the following: (1) low magnesium status is associated with the severity of COVID-19 outcomes, including mortality, and with several disease-related neurological symptoms, including loss of memory, taste, and/or smell; (2) inhaled magnesium as a therapy may improve oxygen status; and (3) magnesium therapy, alone or in combination with zinc, may increase the effectiveness of anti-COVID-19 medications.

**Abstract:**

A large amount of published research points to the interesting concept (hypothesis) that magnesium (Mg) status may have relevance for the outcome of COVID-19 and that Mg could be protective during the COVID disease course. As an essential element, Mg plays basic biochemical, cellular, and physiological roles required for cardiovascular, immunological, respiratory, and neurological functions. Both low serum and dietary Mg have been associated with the severity of COVID-19 outcomes, including mortality; both are also associated with COVID-19 risk factors such as older age, obesity, type 2 diabetes, kidney disease, cardiovascular disease, hypertension, and asthma. In addition, populations with high rates of COVID-19 mortality and hospitalization tend to consume diets high in modern processed foods, which are generally low in Mg. In this review, we review the research to describe and consider the possible impact of Mg and Mg status on COVID-19 showing that (1) serum Mg between 2.19 and 2.26 mg/dL and dietary Mg intakes > 329 mg/day could be protective during the disease course and (2) inhaled Mg may improve oxygenation of hypoxic COVID-19 patients. In spite of such promise, oral Mg for COVID-19 has thus far been studied only in combination with other nutrients. Mg deficiency is involved in the occurrence and aggravation of neuropsychiatric complications of COVID-19, including memory loss, cognition, loss of taste and smell, ataxia, confusion, dizziness, and headache. Potential of zinc and/or Mg as useful for increasing drug therapy effectiveness or reducing adverse effect of anti-COVID-19 drugs is reviewed. Oral Mg trials of patients with COVID-19 are warranted.

## 1. Introduction

The COVID-19 pandemic, caused by the SARS-CoV-2 infection, with 761,402,282 infected individuals and 6,887,000 deaths globally by 19 March 2023 [1], has been a global public health challenge. Therefore, efforts for better understanding of its pathogenesis are needed for the implementation of potential low-cost prophylactic interventions. 

In this regard, the evidence shows that the main surface receptor involved in the SARS-CoV-2 cell entry is angiotensin-converting enzyme 2 (ACE2). However, given that ACE inhibitors or receptor blockers do not modify the virus entrance to cell, the presence of alternative receptors and/or mechanisms has been suggested [2]. Among mechanisms involved in the cell viral invasion are the primary blockade of antiviral innate immunity and the virus’s protection mechanisms against the factors of adaptive immunity [3]. ACE2 exists in two forms: as a transmembrane protein and as a soluble catalytic ectodomain of ACE2. The former undergoes shedding by the tumor necrosis factor (TNF)-α–converting enzyme, in which calmodulin mediates the calcium (Ca)-signaling pathway involved in soluble ACE2 release [4]. A growing body of evidence supports the statement that the ACE2 catalytic ectodomain is the essential entry receptor for SARS-CoV-2 [4], and that recognition of the SARS-CoV-2 spike protein (S) by the ACE2 receptor is needed in the cell invasion stage [3]. 

In this regard, dysfunction of ion channel gradients of the cell membrane has been reported to have a pivotal role in virus entry with the subsequent immune-inflammatory response and development of coagulopathies [5,6]. Given that dysfunction of ion channels is among the mechanisms involved in virus entry in cells, it is plausible to assume that serum electrolyte and trace element imbalances may play a significant role in the development of COVID-19 and mortality from the disease [7]. In this regard, it has been reported that zinc (Zn) participates in both innate and adaptive immune responses and inhibits the activity of the RNA-dependent RNA polymerase of SARS-CoV-2 [8,9]. Sodium modulates the activity of immune cells, and hyponatremia is closely related with pneumonia caused by SARS-CoV-2 [10,11]. Potassium is needed for potential membrane function, and hypokalemia is frequently observed in patients with COVID-19 [12,13]. Ca, a bivalent cation, is required for lymphocyte activation [14] and, undoubtedly, hypocalcemia is a common feature associated with severe disease mortality in COVID-19 [15,16,17,18]. Imbalances or deficiencies of these and other micronutrients can alter bioavailability and/or lead to poor metabolic health, increasing the risk for infection as well as progression to severe disease and complications [19,20]. As an integral part of ATP, and thus cellular energetics [21], magnesium (Mg) is a vital micronutrient in interaction with electrolytes and other essential micronutrients for metabolic balance and health. 

Mg deficiencies, measured by both dietary and blood parameters, widely mimic the risk factors for COVID-19 [22,23,24,25,26]. In this review, we present data on Mg and its potential role in patients with COVID-19.

Mg is an essential trace element that plays a substantial role in physiological, biochemical, and cellular processes [27]. Mg is the second most abundant cation in cells in the body after potassium, with 99% of total-body Mg localized within the intracellular space and approximately 1% in blood and extracellular fluids [28,29].

Mg is involved in all metabolic and biochemical pathways and is required in a large range of vital functions such as bone formation, neuromuscular activity, signaling pathways, bioenergetics, metabolism (of glucose, lipids, and protein), DNA and RNA stability, and cell proliferation and differentiation [30,31,32,33,34,35,36,37]. Enzymatic databases report > 600 enzymes with Mg registered as a cofactor, and there are another 200 in which Mg acts as an activator [38]. However, it must be noted that Mg itself is a substrate rather than a cofactor, as the enzyme substrates are Mg complexes [32,33,34,35].

Mg’s role in suitable immune, vascular, and pulmonary function has been highlighted previously [26,39]. In this regard, Mg is required for the following: proper function of neutrophils and macrophages, cytotoxic activity of T lymphocytes, activation of immunocompetent cells, and inhibition of viral replication [40,41]. It is noteworthy that Mg regulates innate and adaptive immune system activity, which may result in potential protective effects against COVID-19. For instance, Mg stabilizes the membranes of mastocytes, regulates neutrophil and macrophage activity, and inhibits the Toll-like receptor a/nuclear factor-κB (NF-κB) axis [42]. Furthermore, Mg modulates the cytotoxic functions of natural killer (NK) cells and CD8^+^ T lymphocytes [43]. 

Whereas normal Mg levels exert a protective function against viral infection, Mg deficiency may contribute to viral infection. Mg deficiency has been reported to decrease NK and T-cell cytotoxicity, increase NF-κB expression, and exert proinflammatory activity through upregulation of proinflammatory cytokine production in monocytes [44]. Recently, Lötscher et al. [45] demonstrated that extracellular Mg concentration via lymphocyte function-associated antigen LFA-1 regulates CD8 T-cell function. Mg helps T cells attain an active state, transmit signals, reprogram metabolism, form physical bridges to target cells, and ultimately kill errant or infected cells. Mg deficiency primes phagocytes, improves granulocyte oxidative burst, activates endothelial cells, and increases cytokine level synthesis and release (i.e., cytokine storm) [42]. In addition, hypomagnesemia triggers the inflammatory response by activating phagocytic cells, opening Ca channels, and activating NF-κB and *N*-methyl-d-aspartate receptors (NMDARs) [46]. These findings suggest that Mg deficiency may play a critical role in severe outcomes of COVID-19 infection and may explain the increased risk for COVID-19 among patients who are older or have hypertension, obesity, or diabetes, as these individuals usually present with hypomagnesemia and/or low Mg intakes [47,48,49]. 

Given that Mg is compartmentalized such that serum levels represent only 0.8% of the total body store, serum Mg levels may not be an accurate indicator of Mg status [50,51]. However, serum Mg measurement is available in all laboratories and clinical settings, providing an appropriate approach to indicate Mg supplementation if one considers its limitations [51]. We systematically searched the PubMed Medline, Scopus, and Google Scholar databases for studies regarding serum Mg levels and COVID-19, and we present these findings in this review.

## 2. Low Serum Mg and COVID-19

Among areas related to Mg and COVID-19 risk requiring further research is the frequency of hypomagnesemia in subjects with SARS-CoV-2 infection. However, information is scarce [52,53]. Quilliot et al. [52] analyzed serum Mg levels in 300 French patients with COVID-19 at hospital admission. The investigators found that 48% of patients exhibited hypomagnesemia (serum Mg < 1.82 mg/dL [<0.75 mmol/L, <1.5 mEq/L]), including 13% with severe hypomagnesemia (serum Mg < 1.58 mg/dL [<0.65 mmol/L, <1.3 mEq/L]). In a study of 1064 Mexican individuals with COVID-19 at hospital admission, Guerrero et al. [53] found that hypomagnesemia frequency (serum Mg ≤ 1.8 mg/dL) was 44.1%. The hypomagnesemia frequency in both studies was similar, strongly suggesting that the prevalence of hypomagnesemia was high in individuals with COVID-19. Furthermore, hypomagnesemia was associated with infection severity, length of critical care unit stay [52], and death [53], which also suggests that hypomagnesemia is associated with an adverse COVID-19 prognosis (Table 1).

Among tools used to predict the development of severe COVID-19, two studies included serum Mg measurements at hospital admission as a risk factor for adverse prognosis [54,56]. Jia et al. [54] reported that serum Mg levels between 2.19 and 2.26 mg/dL protected from further deterioration. Jia et al. [54] also showed that Mg levels < 2.0 mg/dL are a risk factor for severe COVID-19. In addition, Li et al. [56] assigned a score of 1 to Mg levels < 1.8 mg/dL in the SARS-CoV-ATE risk model for predicting arterial thromboembolism in COVID-19. Among the 16 items on the scale, 14 were assigned a score of 1; only age 40–50 years (2 points) and age > 60 years (4 points) were assigned >1 point. However, the main question that remains to be resolved is whether hypomagnesemia is a previous condition or an early consequence of SARS-CoV-2 infection. Data from the above-mentioned studies were collected at hospital admission for patients with COVID-19 but data on their previous Mg status were not provided. Given that the frequency of hypomagnesemia in the studies by Quilliot et al. [52] and Guerrero et al. [53] was higher than the prevalence of hypomagnesemia in France and Mexico, it is plausible that SARS-CoV-2 infection may cause hypomagnesemia.

Associations of genetically predicted circulating concentrations of 12 micronutrients (i.e., β-carotene, calcium, copper, folate, iron, magnesium, phosphorus, selenium, vitamin B6, vitamin B12, vitamin D, and zinc) with SARS-CoV-2 risk and COVID-19 severity were investigated in a Mendelian randomization study of 87,870 individuals of European descent with a COVID-19 diagnosis and 2,210,804 controls [25]. Significant effects were found only for Mg, indicating that circulating Mg levels in COVID-19 cases were lower relative to the general population. Nonetheless, the relevance of Mg deficiency in increased risk for developing SARS-CoV-2 infection, as well as severe COVID-19, remains to be clarified [26].

### Low Serum Mg and Risk of Developing Severe COVID-19 in Hospitalized Patients

Hypomagnesemia is among the most common metal element deficiencies in hospitalized and critically ill patients and is significantly associated with the need for mechanical ventilation, prolonged stay, and mortality in the intensive care unit (ICU) [57,58]. Hypomagnesemia during hospital stay has been attributed to gastrointestinal abnormalities, renal tubular disease, sequestration in tissues, or the need for diuretic therapy [59]. However, it is important to highlight that at hospital admission, >44% of individuals with COVID-19 have hypomagnesemia [52,53]. Low serum Mg levels are strongly associated with older age [60], diabetes [61], obesity [62], chronic kidney disease [63], hypertension [64], and asthma [65], which are risk factors for adverse COVID-19 outcomes. In this regard, some studies have reported that the above-mentioned risk factors, which in turn are linked to hypomagnesemia, are associated with increased mortality risk in hospitalized patients with COVID-19 [59,66,67,68].

Given that Mg is involved in modulating vascular smooth muscle tone, endothelial function, and myocardial excitability, it is not surprising that hypomagnesemia is associated with the development of cardiovascular disease, hypertension, atherosclerosis, coronary artery disease, congestive heart failure, and cardiac arrhythmias [69]. In this regard, a retrospective cohort study revealed that low serum Mg levels are negatively associated with myocardial damage in patients with COVID-19 [70]. In addition, Mg deficiency may affect bronchial smooth muscle relaxation, causing respiratory dysfunction [71], which can lead to the need for mechanical ventilation, ICU admission, and poor outcomes among patients with COVID-19 [72]. A recent study of patients admitted to the ICU indicated that hypomagnesemia was independently associated with COVID-19 severity (high Acute Physiology and Chronic Health Evaluation [APACHE] score and lung involvement) [55]. Furthermore, hypomagnesemia has been suggested to play a pivotal role in the transition from mild to severe COVID-19 [23]. In particular, patients with severe obesity, diabetes, hypertension, and asthma—conditions related to hypomagnesemia—exhibit high rates of hospitalization and need for mechanical ventilation [73]. Therefore, hypomagnesemia could be considered a poor prognostic marker in hospitalized patients with COVID-19. However, it remains to be determined whether maintaining serum Mg levels at normal ranges could reduce COVID-19 risk. 

Finally, other studies have suggested that high and low serum Mg levels adversely impact COVID-19 outcomes [74,75]. The kidneys regulate Mg homeostasis and thus can be instrumental in hypermagnesemia, and experimental work has also shown that hypermagnesemia can be a consequence of severe experimental dietary Mg deficit [76]. To minimize potential analytic biases that could compromise the conclusions derived from this review, it is important to note that patients with kidney damage were not included. A comprehensive review of hypermagnesemia and COVID-19 is warranted.

## 3. Low Dietary Mg Intake and COVID-19

As with hypomagnesemia, risk factors for COVID-19 correspond to those associated with low dietary Mg intake. Mg intake generally diminishes with aging. Mg deficit is associated with cardiovascular disease, hypertension, type 2 diabetes mellitus, and asthma [77]. In addition, nutritional Mg deficiencies are associated with major chronic noncommunicable diseases such as cardiovascular disease and type 2 diabetes mellitus [47]. In patients with chronic kidney disease, Mg deficiency is common because of restricted Mg intake and impaired Mg reabsorption [78]. Obesity can be considered a profound risk factor for greater COVID-19 susceptibility and severity [79]. The Recommended Dietary Allowance (RDA) for Mg increases with body weight [80]. Thus, persons with obesity have higher dietary Mg requirements than individuals without obesity, making it more likely they will incur a dietary Mg deficit. In addition, low dietary Mg is associated with inflammation, especially C-reactive protein (CRP) [55,56,57].

Whole grains, legumes, nuts, fruits, and vegetables are examples of foods with high Mg concentrations. In previous studies, individuals who consumed a plant-based diet showed a 73% lower risk of moderate to severe COVID-19 [81,82]. However, in populations consuming modern processed food diets, inadequate Mg intakes were common [83]. Low dietary Mg intake may lead to low serum Mg concentration, which can be corrected with oral Mg supplementation [84,85]. One hospital-based study evaluated the nutritional status, dietary habits, and nutrients provided to 367 patients with COVID-19 in the hospital [86]. The results showed that patient Mg intake (mean 373 mg Mg/day) was within the RDA and 95% had only mild or moderate COVID-19 [86]. A cross-sectional study showed that higher dietary Mg intake (i.e., >328 mg Mg/day) was inversely associated with COVID-19 severity and symptoms, with Mg intakes >379 mg/day being even more protective [87]. However, in a cross-sectional study that involved healthy pregnant women who experienced changes in dietary habits and physical activity due to the COVID-19 pandemic, mean dietary Mg intakes were low (<225 mg Mg/day) in all groups (*n* = 168) [88]. In one study of patients with COVID-19 in a general hospital converted to a “COVID hospital”, many changes were made to boost their energy and protein intake after Mg provision in the revised 28-day menu increased from 237 to 295 mg Mg/day [89].

Handwashing and vaccinations help reduce the spread and impact of infections. Nevertheless, nutrition (including Mg) plays an important and complementary role in immune system support [90]. Strong evidence shows that adequate Mg plays a role in reducing inflammation and disease burden among patients with COVID-19 [91]. Mg intake has been shown to be inversely associated with CRP, interleukin (IL)-6, and TNF-α-R2 [92]. To maintain host immunity, adequate intake of dietary agents including Mg is required [93]. Zinc, calcium, iron, and magnesium have important roles in boosting host system immunity and assist in the development and function of lymphocytes, cytokines, free radicals, inflammatory mediators, and endothelial cells [94]. Some micronutrients, including Mg, have been shown to enhance immune system support to fight respiratory infection; therefore, Mg has a vital role in antiviral defense in patients with COVID-19 and may affect the severity of infection, symptoms, and outcomes [95]. Mg is beneficial in reducing the risk of chronic pulmonary disease and viral infection [96], an area of special import for patients with COVID-19.

Vitamin D status is shown to predict COVID-19 risk, infection severity, and death [67,97]. Serum Mg levels are associated with serum vitamin D levels [98]. Mg status is important in vitamin D metabolism, with Mg being required for proper vitamin D activation and metabolism [99]. Mg deficiency may impair vitamin D metabolism [100], whereas oral Mg supplementation has been shown to increase serum vitamin D levels [101]. In addition, lower Mg intakes have been shown to increase the risk of death in middle-aged and older men with low serum 25(OH)D_3_ concentrations [102]. Finally, oral Mg supplements, along with vitamins D and B12, forestalled oxygen therapy and/or ICU support of patients [103].

Consumption of high-Mg drinking water has long been associated with lower risks of cardiovascular death [104]. In addition, US researchers found that COVID-19 infection risk during early transmission was greater in populations in areas low in Mg [105]. The main mechanisms of the effects of drinking mineral water on the rehabilitation of new coronavirus infection convalescents are nonspecific hormone-stimulating effects in the form of pronounced activation of the gastroenteropancreatic endocrine system, which is capable of integrating substance and energy exchange following the current needs of the body and also excreting vasoactive factors modulating vital functional system activity. Intake of mineral water with a high content of hydrocarbonate ions, magnesium, sodium, and carbon dioxide saturation with general water mineralization from between 5 and 6 to between 11 and 13 g/L has the maximum effect [106]. As a result of food refining and processing, the Western diet is often low in Mg [107]. Populations consuming “traditional diets” show mean Mg intakes of 450–700 mg/day among adults, while those consuming “modern processed food diets” consume mean intakes of 250 and 350 Mg/day [83] (Figure 1). Whole grains are among the foods highest in Mg content; however, when refined, whole grains such as wheat, rice, and maize lose between 75% and 85% of their Mg [108]. We speculate that populations consuming diets with >50% cereal grain food processing are at risk of potential Mg deficit and thus higher rates of and more severe COVID-19 infection. For example, Malawi supplies of Mg are adequate for >80% of households [109]. According to World Health Organization data, Africa (an area with low processed food consumption) has the lowest number of confirmed deaths from COVID-19 compared with Europe, the Americas, the Western Pacific, Southeast Asia, and the Eastern Mediterranean [1] (areas with high processed food consumption). The impact of dietary Mg intake and status on COVID-19 is a promising area for future study.

## 4. Mg Supplements and COVID-19

The role of Mg supplements in preventing or treating chronic disorders related to the respiratory system (asthma), reproductive system (preeclampsia), nervous system (migraine), digestive system (constipation), cardiovascular system (hypertension), and endocrine system (diabetes) has been shown previously [110], just as oral Mg supplements been shown to lower serum CRP [111]. After the global COVID-19 outbreak, scientists focused on approaches to prevent or treat this infectious disease. They also strove to introduce supplements or medications to reduce COVID-19 symptoms in patients [112]. Essential supplements to modulate the immune system and interferon (IFN) signaling pathway (e.g., vitamin D, Zn, and Mg) were offered in this regard [113]. Mg was highlighted as an element involved in the immune and inflammatory pathways of COVID-19 [24]. Accordingly, Mg supplements were suggested to prevent and treat COVID-19 [110]. In a previous study, a short course of a Mg supplement in combination with vitamins B12 and D was administered to 17 patients with COVID-19 [103]. These patients showed decreased requirements for oxygen therapy and ICU support compared with controls [103]; however, serum Mg levels were not measured and reported in this study. Mg supplements may decrease symptoms in patients with hypomagnesemia, whereas they may not be helpful for patients with normal serum Mg levels [114]. Administration of Mg supplements may restore intracellular Mg, which leads to regulation of the cytotoxic functions of NK and CD8^+^ T cells and reduction in cytokine overproduction [115]. There are two important issues to discuss when recommending Mg supplements as a supportive treatment for patients with COVID-19: (1) measurement of serum Mg levels and (2) bioavailability of Mg supplements. Ionized serum Mg (iMg) or total serum Mg (tMg) can be measured to assess Mg status. Although iMg may better predict clinical outcomes, especially in critically ill patients, it cannot be measured in many clinical settings because specialized equipment is required [116]. In a previous study, Zhan et al. [117] compared the efficacy of iMg whole blood concentrations with tMg concentrations to assess responses to an oral Mg supplement (MgCl_2_). The results showed significant increases in iMg concentrations compared with tMg following oral MgCl_2_ administration. The authors concluded that the measurement of iMg whole blood concentration is more sensitive than tMg when a Mg supplement is administered [117]. However, a reliable reference range for iMg should be available in order to compare the results of various studies.

The bioavailability of Mg supplements varies within a broad range and depends on various factors. The type of salt and the formulation of Mg supplements are two known factors that may affect the absorption rate. Some studies have shown that the bioavailability of organic Mg salts (lactate, aspartate, amino acid chelate, and citrate) is slightly higher than inorganic products; however, other studies have not confirmed these results [28]. The different study designs on the bioavailability of Mg salts make it difficult to predict which type of Mg supplement has a better absorption rate. Solubilized Mg formulations (e.g., effervescent tablets) have more bioavailability than slow-release formulations [118], and it is better to recommend these to patients who need a rapid increase in Mg serum levels. Drug-drug and drug-food interactions are also important factors influencing the absorption and bioavailability of Mg supplements. Mg absorption is pH dependent and mostly occurs in the small intestine via a passive pathway [119]. Changing gastric acid secretions and intestinal pH with drugs such as proton-pump inhibitors can decrease Mg absorption [120]. High doses of other minerals (e.g., calcium, iron, and zinc) also decrease the absorption of Mg supplements by competing at absorption sites. In addition, Mg chelation with other medications (e.g., fluoroquinolones) and food contents (e.g., oxalic acid) impairs gastrointestinal absorption of Mg supplements [28,120].

Due to variations in Mg supplement bioavailability and serum Mg concentrations, study designs on the beneficial effects of Mg supplements in patients with COVID-19 should be based on precise standards. Defining these standards to interpret and compare the results is recommended.

## 5. Mechanisms of Action of Mg in Pulmonary Complications of COVID-19

The respiratory system is the main organ involved in COVID-19, and hypoxia is a significant cause of morbidity and mortality resulting from this disease. According to previous studies, the primary cause of hypoxia in COVID-19 is ventilation–perfusion (VQ) mismatch [121,122]. The phenomenon of VQ mismatch occurs in the alveolar-capillary unit due to airflow incompatibility in the alveoli and pulmonary capillary blood flow around these alveoli. Some lung areas, such as nearly normal ventilated alveoli with microvascular thrombosis and vasoconstriction, have high V/Q. In other parts of the lung, blood flow is diverted to dilated vessels around poorly ventilated alveoli, which show low V/Q. Therefore, these events lead to VQ mismatch; awareness of this mechanism is helpful for determining therapeutic solutions [123].

To improve ventilation and perfusion and to reduce VQ mismatch in patients with severe COVID-19, nebulization of several agents (e.g., prostacyclin analogues, nitric oxide [NO], and Mg sulfate) has been proposed [124,125,126]. Among these medications, Mg sulfate is a more accessible and inexpensive compound with multiple effects to improve the oxygenation of patients with COVID-19. Mg sulfate has anti-inflammatory, bronchodilatory, vasodilatory, and antithrombotic effects [127,128]. Treatment with inhaled Mg sulfate may play the best role in improving the oxygenation of hypoxic patients with COVID-19 due to better lung accumulation with direct effects. In this regard, inhalation of Mg sulfate causes bronchodilation and improves ventilation; the accumulation of Mg sulfate in well-ventilated alveoli then causes vasodilation and increases perfusion in the capillaries around these alveoli. As a result, increased perfusion in well-ventilated alveoli could reduce VQ mismatch and improve oxygenation [126]. Based on this theory, a multicenter, open-label, randomized controlled trial was conducted in Iran during 2020 and 2021 to investigate the effect of inhaled Mg sulfate in hospitalized patients with COVID-19 [126]. In the intervention group, patients inhaled 50 cc of Mg sulfate (2%) every 8 h for 5 days in addition to standard treatment. In the control group, only standard treatment was received according to Iran’s national guidelines. Within 5 days, patients’ respiratory symptoms and oxygenation were checked. This study (manuscript in preparation) was conducted in hospitalized patients who were severely or critically ill with COVID-19 [126]. These findings suggest that Mg inhalation may have a better effect on oxygenation for patients with severe disease and hypoxia mainly caused by VQ mismatch compared with critically ill and very severely ill patients who often have some degree of intrapulmonary shunt (Table 2). The bronchodilatory effect of nebulized Mg has been investigated in the treatment of asthma [129]. In our recent study, the vasodilatory role of Mg, in addition to its bronchodilatory effects, was investigated. Moreover, the anti-inflammatory and antithrombotic effects of inhaled Mg sulfate may also be useful in improving oxygenation. One may hypothesize that increased anti-inflammatory and antithrombotic actions can be induced by the direct effect of Mg sulfate inhalation on the lung parenchyma and capillaries. To prove this theory, molecular and cellular pathology studies should be designed.

## 6. Mg in Neurological and Psychiatric Complication of COVID-19

COVID-19 infection has many complications that manifest in patients of all ages, including strong effects on the central nervous system (CNS) and peripheral nervous system [132,133]. Neurological and psychiatric complications of the disease are among the most common and sometimes have severe forms [134]. Some of the complications manifest only during hospitalization, whereas others appear or continue after hospital discharge [135]—sometimes long after the patient is declared healthy and discharged from the hospital. Neurological and psychiatric complications have a major impact on the evolution of the health status of these patients [136]. The essential mechanism in producing all of these complications is the existence of SARS-CoV-2 receptors in the CNS and binding of the virus to these receptors.

The most frequent neurological and psychiatric complications are headache, dizziness, convulsions, and asthenia [94,137]. Another common neurological disorder (sometimes considered a symptom and sometimes a complication of the disease) is loss of taste and smell (olfactory and/or gustatory dysfunction), which occurs in many COVID-19 cases. Disorientation, ataxia, confusion, loss of consciousness, cranial nerve deficit, Guillain-Barré syndrome, convulsions, and hallucination are much less common [137,138].

With regard to neurological and psychiatric disorders that appear in patients during the COVID-19 course, it is difficult to separate the disease symptoms from its complications. However, symptoms that appear or persist after the patient has been considered cured, is no longer receiving anti-COVID-19 treatment, and is discharged must also be considered complications. In addition to the complications listed earlier, it is important to mention those that appear in the offspring of women with COVID-19 during pregnancy. Some of these children have been diagnosed with autism, epilepsy, and even schizophrenia [139]. In some cases of COVID-19, encephalitis (a severe complication) has been observed [140,141]. Implications of Mg in the pathogenesis and evolution of these CNS complications of COVID-19 are multiple, as described next.

### 6.1. Guillain-Barré Syndrome and Encephalopathies

Guillain-Barré syndrome is a very severe acute paralytic neuropathy. This syndrome is the most frequent paralytic neuropathy worldwide [142]. Guillain-Barré syndrome has been reported as a complication of COVID-19 [137] that persists long after recovery from the disease [143]. About 25% of these patients develop not only rapidly progressive weakness of the extremities but also respiratory insufficiency [144]. This impairment is difficult to differentiate from the respiratory insufficiency produced by COVID-19 through its direct action in the respiratory system.

Acute flaccid paralysis is essentially an acute inflammatory neuropathic disease [145]. The mechanisms by which Mg can reduce disease severity include anti-inflammatory action, cytokine storm reduction when it occurs, and protective action at the myelin of the peripheral nerves.

Cerebral hypoxia and the genesis of acid metabolites through the increase in anaerobic metabolism are also involved in the occurrence of toxic encephalopathy in some patients with COVID-19 [133]. Encephalopathy degrades patients’ mental states to different degrees, varying from patient to patient. Severe forms of necrotizing encephalitis are considered to be produced by the cytokine storm. By reducing this storm, Mg suppresses one of the essential mechanisms of encephalopathies.

### 6.2. Memory and Cognition

Memory and cognition are complex multifactorial processes in which there is a complicated relationship between genetic and nongenetic factors [146]. Cognition and memory disorders have been observed in patients with COVID-19 [147,148]. Accurate quantification of severity and frequency is difficult because even if these disorders are observed during hospitalization, it is only sometimes known whether the patient presented them before SARS-CoV-2 infection. Mg plays a major role in the mechanisms of memory and cognition, and Mg deficiency is involved in a significant reduction in memory [149,150]. Restoring the normal tissue and/or circulating concentration of Mg determines a slow recovery of memory. We hypothesize that a low Mg level (found in many individuals with COVID-19, but not in all) is involved in the memory deficit found in these patients.

The mechanisms by which Mg is involved in the prevention of memory loss or in the recovery of memory in patients in whom it has been diminished are as follows:increased neuroplasticity [151],upregulation of cAMP response element-binding protein (CREB)-mediated transcription,modulation of the activity of some transcription factors (c-Fos, nuclear factor-κB [NF-κB]),increased brain-derived neurotrophic factor (BDNF) action,reduced oxidative stress in the brain,increased reduced glutathione levels, orreduced proinflammatory cytokine synthesis and release [23].

Activation of the c-Fos gene is necessary for memory formation [152]. Mg activates this gene, whereas experimental Mg depletion impairs c-Fos activity.

Mg concentrations modulate the activity of the proto-oncogene c-Fos and NF-κB. A low Mg level upregulates the activity of these transcription factors. Mg administration, Ca channel blocking with nifedipine, or Ca removal from the incubation medium leads to the return to normal activity of these transcription factors [153].

Mg administration has been shown to improve memory deficits and reverse upregulation of TNF-α/NF-κB signaling in the hippocampus [154]. A normal level of reduced glutathione is beneficial for memory and other brain functions [155]. Due to increased oxidative stress and a high level of proinflammatory cytokines in patients with COVID-19, the glutathione level decreases in the brain. In experimental studies, short-term spatial memory is the most affected by reduced glutathione depletion. Mg increases reduced glutathione levels and thus slows memory decline, respectively, which contributes to faster restoration of affected memory [155].

Memory is mediated by neuroplasticity and synaptic interconnection [156]. Mg improves synaptic plasticity, especially in the hippocampus. In neuron cultures, a good correlation has been shown between the increase in synaptic plasticity and the extracellular concentration of Mg.

In experimental studies in rats, increasing brain Mg^2+^ concentration is positively correlated with increased working, long-term, and short-term memory. Experimental data show that reducing Ca^2+^ influx in neurons increases neuroplasticity. Mg is a partial antagonist of Ca entry into the neuroglia and neurons and, in this way, increases neuroplasticity [151,157].

CREB-mediated transcription is involved both in the mechanism of long-term memory and in short-term memory [152]. Mg upregulates CREB-mediated transcription, meaning that it is involved in both the mechanism of long-term and short-term memory [158]. Deficiency in this biometal in mice induces impairments in hippocampus-dependent memory.

Abnormal functioning of the glutamatergic system through abnormally intense stimulation of NMDARs by glutamate is also involved in memory disturbances. Synaptic plasticity is important not only for memory but also for cognition. Administration of Mg and restoration of normal concentrations of this cation improves both memory and cognition [159]. Unfortunately, in the context of problems related to hospital care of patients with COVID-19, the cognitive capacities of these individuals are rarely evaluated.

SARS-CoV-2 infection depletes intracellular adenosine triphosphate (ATP) reserves, among other effects, and consequently reduces cellular energy metabolism. To restore these reserves, Mg is strictly necessary [160]. Chronic Mg deficit enhances oxidative stress and depresses antioxidant defense [161].

ATP is the cell’s main source of energy. Mg plays an essential role as a cofactor for ATP [33]. In this way, a decrease in Mg concentration determines the decrease in cellular energy at the neuronal level, with important consequences in brain activity, including memory impairment.

BDNF is also important for memory and learning. An increase in BDNF concentration in the hippocampus determines improvement in cognitive function [162]. Some authors consider BDNF as one of the most important molecules involved in memory and cognition [163]. Mg increases the BDNF level in the brain and thus improves memory [164,165]. For unknown reasons, a decrease in extracellular Mg has not been shown to affect amygdala-dependent memory [166].

### 6.3. Taste and Gustatory Dysfunction, Loss of Smell, and Loss of Appetite

The molecular mechanism of taste sensation is not completely known. Experimental studies have shown that the TAS2R7 taste receptor is activated by MgCl_2_ (and by MnCl_2_ and ZnSO_4_) [167]. There are other taste receptors (TAS2R14, TAS2R10, TAS2R38, and TAS2R16) but they are not activated by Mg or Zn salts. Reduced Mg in patients with COVID-19 can cause a decrease in, change in, or complete loss of taste [168].

Loss of smell is common in patients with COVID-19, both in hospitalized patients and in those with mild forms of the disease that do not require hospitalization. Restoration of the acuity of the olfactory senses occurs mainly under the action of Mg and Zn [169]. A low Mg level contributes to the loss of smell in patients with COVID-19 [168,170].

Mg deficiency also causes loss of appetite [171], which is frequently encountered in both hospitalized patients [86] and outpatients [172] with COVID-19.

### 6.4. Ataxia

Among various neurological diseases in which their pathogenesis involves hypomagnesemia are cerebellar syndromes, which include ataxia [25]. In a previous study, Mg administration contributed to the improvement of cerebellar clinical manifestations in these patients [173]. Mg administration also rapidly improved the ataxic manifestations [174,175].

Severe chronic Mg depletion produces a decrease in some cerebellar functions. This also occurs in some patients with COVID-19 [176,177]. There are no clinical data showing the effect of Mg administration in patients with COVID-19 who present with ataxia, but we consider that Mg administration should be considered an important component of treatment for these patients.

### 6.5. Confusion, Delirum, and Consciousness Disturbances

One clinical manifestation of hypomagnesemia is the presence of confusional states and, more rarely, delirium [25]. Hypomagnesemia is one of the electrolyte disorders that are sometimes associated with the appearance of delirium [178]. The problem of disturbed consciousness is very complex because it can evolve from dysphoria and disorientation to loss of consciousness [133]. Impaired consciousness occurs in about 14% of patients with COVID-19 [132,138]. Because the exact mechanism of the production and maintenance of consciousness is unknown, it is difficult to indicate where Mg is involved. Both hypomagnesemia and hypermagnesemia sometimes cause severe disturbances of consciousness [179,180]. Hypomagnesemia is common in patients with COVID-19, but there are no consistent data regarding the presence of significant hypermagnesemia in these patients.

Mental fatigue and trouble with attention have been reported in patients with COVID-19 [181,182]. These clinical manifestations were reversible when Mg was administered [183].

### 6.6. Cranial Nerve Deficits and Cranial Nerve Palsy

Peripheral autonomic nervous system disorders are due to the direct action of the virus on peripheral nervous structures and occur in about 2.5% of hospitalized patients. Demyelination, cranial nerve palsy, and axonal neuropathies are present in some patients with COVID-19 [184]. Complications of this disease include damage to the cranial nerves, especially lesions of the bulbar cranial nerves [185].

A direct consequence of this damage is the involvement of cranial nerve impairment in oropharyngeal dysphagia after COVID-19. After the intubation of some patients with COVID-19, neuropathy of some cranial nerves and dysphagia have also been found [186,187]. This complication is quite rarely reported. Mg, which has a protective action at the level of myelin, could reduce this complication.

Some cases of bilateral palsy of the vocal cords after COVID-19 infection have also been reported. There are no data regarding the involvement of Mg in these complications.

### 6.7. Convulsions, Child Epilepsy, and Hallucinations

In patients with COVID-19, convulsions occur mainly due to encephalic inflammation and hypoxia, which lower the convulsive threshold. Some patients experience focal seizures, whereas others have generalized seizures [188]. These seizures can be tonic or tonic-clonic and appear in children and in some adults. The mechanisms of producing convulsions are different [189]. Hypomagnesemia is associated with the production of convulsions. Lack of Mg lowers the convulsive threshold, increases the glutamate concentration in the brain, and reduces the action of inhibitory GABAergic systems [183]. Recurrent episodes of confusion have been reported in hypomagnesemia in various cases, such as those necessitating proton-pump inhibitor use to treat gastric and duodenal ulcers [190].

To our knowledge, no systematic investigation has been conducted in patients with COVID-19 to evaluate the correlation between plasma Mg concentration and the occurrence of hallucinations. In other pathological situations such as patients with some forms of schizophrenia or individuals with neoplastic conditions in terminal stages receiving chemotherapy, the association of hallucinations with hypomagnesemia has been reported [191].

### 6.8. Demyelination and Axonal Neuropathies

Both genetic and epigenetic factors are involved in the mechanism of producing demyelination [192]. One of those in the last category is Mg deficiency.

One disease in which a significant Mg deficit is evident is multiple sclerosis. Demyelination plays an essential pathogenic role in the progression and evolution of this disease. In patients with multiple sclerosis, the most marked reduction in Mg^2+^ content has been observed in the CNS white matter and demyelinated plaques. Mg deficiency is involved in nerve dysfunction and demyelination [193,194]. This reduction in Mg concentration at the level of demyelinated areas and multiple sclerosis plaques is associated with general hypomagnesemia in these patients [195]. In other CNS diseases with demyelination, low Mg levels have also been reported. Mg has a stabilizing effect on myelin. Mg administration to patients with multiple sclerosis has been shown to decrease the frequency of relapse [196]. Demyelination from multiple sclerosis is associated not only with a low level of plasma Mg but also with a reduction in the concentration of this element in the cerebral spinal fluid [197]. Myelin is sensitive to the oxidative degradation produced by excess free radicals. Mg and Zn reduce oxidative stress and thus could have a partial protective effect against the demyelination sometimes encountered in COVID-19. In the blood of patients with multiple sclerosis, low Mg levels are associated with a decrease in the concentration of other metallic elements (chromium, cobalt, zinc, and others) [198]. These data contributed to the hypothesis that the involvement of biometals in the demyelination process also found in patients with COVID-19 is complex [192]. Existing data on the concentration of metal trace elements in patients with COVID-19 are few, and correlations between the concentrations of these trace elements and Mg are lacking. We consider that the partial protective mechanism of Mg in multiple sclerosis is the same as the mechanism by which Mg can reduce demyelination in the case of COVID-19 complications.

### 6.9. Headache and Dizziness

Headache is common in patients with COVID-19 and has been reported in numerous studies. For example, headache was reported in 11–34% of hospitalized patients with COVID-19 [130,138]. Previous studies suggest that the main pathogenic mechanism is the entry of the virus into the CNS and the increased synthesis and release of cytokines [199].

After SARS-CoV-2 fixes on the receptors at olfactory mucosa by transsynaptic migration through the olfactory route from the nasal cavity, the virus affects both the trigeminal branches and the trigeminal ganglion [200].

The following are mechanisms by which SARS-CoV-2 infection can produce headache:by increasing the amount of circulating proinflammatory cytokines (Il-1β, TNF-α, and others) that enhance trigeminal nociception [201];through direct action of the virus on trigeminal nerve endings;by local hypoxic phenomena that also affect the peripheral trigeminal endings [130]; andby increasing oxidative stress and free radical formation.

Headache reduction by Mg is achieved through several mechanisms:by reducing cerebral and pericranial vascular smooth muscle spasms, modulating the NO level in the cell, and eliminating NO trapped inside the cell [202]; andby inhibiting IL-1β and TNF synthesis and reducing neuroinflammation.

Mg reduces oxidative stress, free radical formation, and hypoxia through vasodilatation. All of these findings indicate that Mg can reduce headache produced by SARS-CoV-2 infection.

In the case of Gitelman syndrome, in which healthy patients present with dizziness and headache, Mg administration has been shown to improve symptoms and alleviate dizziness [203,204]. Regarding patients with COVID-19, there are currently no clinical studies regarding the therapeutic effect of Mg to improve dizziness.

### 6.10. Immunity

Mg deficiency is involved in the pathogenesis of COVID-19 complications due to decreased intracellular and extracellular Mg that leads to decreased antiviral immunity [205]. Although hypomagnesemia is involved in the occurrence of some COVID-19 complications [115], there are no data to show a therapeutic benefit of hypermagnesemia in this disease [206].

Hypomagnesemia is statistically significantly associated with increased mortality of patients with COVID-19. The causes of death among these patients are multiple, but CNS dysfunctions are also involved (without the existing data to identify the frequency of these involvements) [68].

A potentially important consideration is the ratio of Mg and Ca concentrations in patients with COVID-19. There are little data in this regard, but some studies show that a weight ratio of Mg:Ca serum concentrations ≤ 0.2 (or molar Mg/Ca serum ratio ≤ 0.33) determines the occurrence of serious complications and is strongly associated with mortality in patients with severe COVID-19 [53]. Unfortunately, there are no studies showing the importance of the Ca/Mg ratio in the occurrence of neuropsychiatric complications of COVID-19 or regarding their severity.

The results of these studies suggest that Mg is not only a supportive treatment in COVID-19 [110], but it can also directly contribute to healing or improvement of some neurological and psychiatric complications of the disease (Table 3).

## 7. Interrelationships between Mg, Zn, and Agents Used to Treat COVID-19

Several agents have been used or are still used today to treat COVID-19. Some examples include ivermectin, azithromycin, chloroquine, hydroxychloroquine, casirivimab, dexamethasone, imdevimab, sotrovimab, tocilizumab, remdesivir, amantadine, moxifloxacin, mefloquine, molnupiravir, anticoagulants, favipiravir, and others [207,208,209,210,211,212]. Important interrelationships with Mg and Zn—two of the most important biometals in the human body—are known for some of these agents. With regard to their use in COVID-19 treatment, the interrelationships between these medications and two of the main electrolytes in the human body, Mg and Zn, are complex. The four groups of interrelationships are as follows:pharmacodynamic interactions between Mg^2+^ and Zn^2+^ and the action of anti-COVID-19 drugs, including a) direct influence on the drug mechanism of action and b) indirect influence through the influence on the body immunity of patients with COVID-19;pharmacokinetic interactions (related to the influence of Mg and Zn on the absorption, transport in the blood, and elimination from the body) of the drugs used;influence of anti-COVID-19 medication on the plasma or tissue concentration of these two cations; andinfluence of Mg^2+^ and Zn^2+^ on the adverse effects of anti-COVID-19 medication.

Mg and Zn can indirectly influence the efficacy of anti-COVID-19 drugs by reducing the intensity and frequency of some adverse effects and by their involvement in the immune response, which, together with the action of these drugs, plays an essential role in the evolution and prognosis of patients with COVID-19. In evaluating the interrelationships of Mg^2^, Zn^2+^, and anti-COVID-19 drugs, the following factors must also be considered:5.Among the numerous drugs that are or have been used to treat COVID-19, these interrelations are partially known only for some. Future studies are needed.6.Many drugs that have been or are used in anti-COVID-19 therapy are also used to treat other diseases. Their pharmacokinetic characteristics and mechanism of action remain the same and are intrinsically determined by their molecular structure. Some pharmacokinetic characteristics could be modified if new pharmaceutical forms are used.7.Frequently, patients with COVID-19 receive not only anti-COVID-19 therapy but also treatment for preexisting chronic diseases. Drugs used to treat these conditions can change Mg and Zn plasma concentrations and thus indirectly influence anti-COVID-19 therapy.8.Extracellular and intracellular Mg levels and Zn plasma levels in hospitalized patients with COVID-19 are highly variable. Many patients have hypomagnesemia or hypozincemia (or both) upon admission, and many develop imbalances of these elements during the disease course.9.Determinations of serum Mg and Zn concentrations at admission and during hospitalization are inconsistent (and in many cases, only sporadically undertaken).10.Oral nutrition of these patients can be deficient, and the solutions administered parenterally rarely aim to correct Mg deficiency.

Patients with COVID-19 receive anti-COVID-19 treatment (usually made up of several associated medications) and medications used to treat any preexisting diseases. With all of these medications, there can be interactions of various types independent of interactions of the anti-COVID-19 medication with Mg and Zn.

Mg and Zn homeostasis is important [26], but all of the factors shown previously affect pharmacotherapy against COVID-19. Disturbances of the hydroelectrolyte balance during hospitalization of patients with COVID-19 have been observed not only for Mg and Zn but also for other electrolytes. Sodium and potassium serum concentrations are significantly lower in these patients compared with healthy individuals of the same age [213]. Serum concentrations of the main electrolytes must be determined for all patients, both at admission and during treatment.

### 7.1. Pharmacodynamic Interactions

Dexamethasone is the most frequently used glucocorticosteroid in the treatment of patients with COVID-19. Experimental studies have shown that coadministration of dexamethasone with Mg significantly decreased IL-13 concentration in bronchoalveolar lavage fluid and reduced immunoglobulin E concentration in mice sensitized with ovalbumin compared with animals that received only dexamethasone [214]. Yet there is another type of pharmacodynamic interaction between Mg and glucocorticosteroids. In the case of experimental models of neuroinflammation and damage to the blood-brain barrier (BBB) by TNF-α administration, dexamethasone administered in combination with Mg preserved BBB integrity. Because Mg alone does not have this effect, it is likely attributable to the potentiation of the effects of dexamethasone by Mg [215]. 

Studies show that Zn increases the effectiveness of ivermectin and ribavirin therapy regarding viral clearance rates determined with a nasopharyngeal swab. In a clinical trial, clearance of the nasopharyngeal swab was 0% 7 days after the start of treatment in the group who received only the anti-COVID-19 medication and 58% in the group that received Zn and two anti-COVID-19 drugs. Clearance was 13.7% and 73%, respectively, 15 days after the start of the therapy. These data show that Zn has a potentiating effect on some anti-COVID-19 drugs regarding viral clearance in nasopharyngeal swabs [216]. Randomized clinical studies have shown that Zn and ivermectin have synergistic actions among moderately symptomatic patients with COVID-19 [217]. Apart from this synergism, Zn administered alone (oral Zn, 20 mg/8 h) inhibits the replication of many viruses (including SARS-CoV-2). This fact suggests that Zn should be administered immediately after viral infection has occurred in order to avoid severe forms by partially blocking viral multiplication of COVID-19. 

By reducing cytokine synthesis and oxidative stress [218], Zn interferes with the pathogenic mechanism of SARS-CoV-2 infection. Another mechanism by which Zn reduces the effects of viral aggression on the body is the reduction in apoptosis by inhibiting caspase-3 (which is an apoptotic protease) [219]. The combination of Zn with anti-COVID-19 medication has also been shown to significantly reduce mortality. A study conducted with 3453 patients (median age, 64 years) showed that administration of Zn^2+^ ionophore was associated with a 24% reduced risk of in-hospital mortality [220]. Another study showed that Zn sulfate and with a Zn ionophore added to anti-COVID-19 therapy with azithromycin and hydroxychloroquine significantly reduced mortality [221], reduced symptom severity, and also increased quality of life of these patients [222,223]. Clinical and experimental data indicate that Zn administered with anti-COVID-19 therapy is associated with patient benefit. To increase effectiveness, Zn (which is a widely localized predominantly extracellular element) must be combined with an ionophore that introduces this cation into the cell and increases the intracellular Zn concentration. Zn(^2+^) plus an ionophore for Zn (e.g., pyrithione) inhibits SARS-CoV-2 replication in cell cultures [224]. 

The administration of anticoagulants (heparin, enoxaparin, and others) is considered an approach to reduce mortality in severe cases of COVID-19. Mg sulfate and other Mg compounds also have anticoagulant action [225,226]. In addition, the Mg^(2+)^ cation reduces platelet aggregation. These two actions indicate that the combination of Mg with anticoagulants administered parenterally in patients with COVID-19 is beneficial. Mg has a synergistic action with heparins and can reduce thrombus formation (an important cause of mortality in patients with COVID-19). The anticoagulant effect of Mg is not intense and is not comparable to the effect of heparins; but certainly, the combination of the two effects is beneficial to patients. In addition to its anticoagulant action, heparin has some antiviral actions. The formation of a complex between Mg and heparin causes conformational changes in the heparin molecule. Following these changes, the antiviral action of heparin increases significantly against some viruses such as human adenovirus and herpes simplex virus type 1 [227]. Replication of these viruses is reduced by the action of this complex. There are no data on the action of the heparin-Mg complex on SARS-CoV-2 replication, but it is possible that part of the favorable effect given by the administration of heparin to patients with COVID-19 is also due to this action. These data show, from another point of view, the importance of Mg plasma concentration in these patients.

Platelet aggregation factor (PAF; glyceryl-ether phospholipid (1-*O*-alkyl-2-acetyl-*sn*-glycero-3-phosphocholine) is a lipid factor synthesized to a large extent in platelets and has powerful platelet-aggregating [228,229] and proinflammatory actions. PAF also has strong bronchoconstriction action [230,231]. Another mechanism that can explain the beneficial effect of the combination of Mg and Zn with anti-COVID-19 medication is the inhibitory effect of these cations on the synthesis and action of PAF [232]. Mg and Zn inhibit PAF synthesis and action [183,233]. Zn chelate complexes and Mg chelate complexes have PAF inhibitory activity mainly attributed to stereochemical interactions. Chelating agents (e.g., Mg^2+^ chelate) reduce the activity of some PAF biosynthetic enzymes such as lyso-PAF-acetyltransferase. Mg and Zn also inhibit other actions of PAF, such as gastric ulcerogenic action [234].

PAF could increase SARS-CoV-2 infection severity by stimulating ACE and ACE2 activity [235]. This enzyme is a receptor for SARS-CoV-2 [236,237] and is found in many organs, including at the epithelial level in the lung. After becoming fixed to this receptor, the virus can penetrate cells and produce the disease. Inhibition of this enzyme has therapeutic potential in the treatment of COVID-19. There are two ACEs (both being carboxypeptidases) in the human body. The first (ACE) transforms the inactive angiotensin into the active form (with eight amino acids), which causes vasoconstriction, and has proinflammatory and other actions. On the contrary, ACE2 transforms angiotensin II into a polypeptide with seven amino acids (Ang1–Ang7), which has anti-inflammatory and vasodilatory action. ACE2 activity depends on Zn. A normal level of serum Zn favors ACE2 activity. This enzyme is a receptor for SARS-CoV-2 both in the presence and absence of Zn, but its beneficial actions (anti-inflammatory, vasodilatory, and others) require a normal Zn concentration [238]. In this way, Zn combined with anti-COVID-19 therapy can improve disease symptoms. Angiotensin II receptor blockers have also been proposed for anti-COVID-19 therapy [239]. Because ACE2 is a primary receptor of SARS-CoV-2 [240], PAF could facilitate the fixation of the virus on the cell surface. By inhibiting PAF synthesis and action, Mg and Zn could indirectly reduce SARS-CoV-2 entry into cells. It is debatable whether some PAF receptor antagonists, such as ginkgolide A from *Gingko biloba* [241], could increase the effectiveness of anti-COVID-19 therapy.

Zn sulfate together with a Zn ionophore significantly reduced mortality [221] and symptom severity [222,223] compared with anti-COVID-19 therapy with azithromycin and hydroxychloroquine. Because patients hospitalized with COVID-19 have numerous nutritional deficiencies (partly acquired before and partly accumulated during the disease) and because their nutritional intake is disturbed, it is necessary to evaluate the nutritional status of all individuals hospitalized with the disease [242]. Determination of plasma concentrations of Mg and Zn at admission is especially important since data show that about 2 billion people consume a diet deficient in these cations [47,243]. This evaluation is more difficult to achieve in the case of patients with COVID-19 who are not hospitalized. All nutritional deficiencies must be quickly corrected. We believe that this would have an important influence on both the evolution of the disease and the effectiveness of treatment. Most patients with COVID-19 are aged >60 years, and most deaths resulting from the disease occur in this age group. Increased cytokine synthesis, cytokine storm, and exacerbation of oxidative stress are involved in the severe disease as well as increased mortality encountered in these patients. Zn inhibits cytokine synthesis and reduces oxidative stress [244], and Mg deficiency has been related to cytokine storm [42,115]. Mefloquine, which has been used in anti-COVID-19 therapy, stimulates the activity of the choline/Mg^2+^ antiport system. It is not known whether this influences on the anti-COVID-19 action of the drug, but it is known that this effect is not related to its antimalarial action [245]. For several viruses, including coronaviruses, Zn reduces entry into the cell via blocking of polyprotein processing or by inhibiting the RdRp RNA-dependent RNA polymerase. In this regard, the effects of Zn differ from those of Mg, which does not have this inhibitory action [224,246]. Zn administration is also important for the correction of taste disorders that are common in patients with COVID-19 [247]. Plasma Zn concentration is significantly lower in older adults and patients with diabetes [248,249], which correlates with the much higher mortality caused by COVID-19 in these patients. Zn has multiple immune implications regarding viral infections and not only in the case of SARS-CoV-2 infection [250,251]. This biometal plays an important role (considered critical by some authors) in antiviral immunity [252,253]. Besides the other implications of this cation, this could be the reason why the circulatory Zn concentration (as well as Mg) should be determined immediately after admission and why Zn salts should be administered, in many cases, to correct deficiency.

Zn administration increases antiviral immunity (including immunity in SARS-CoV-2 infection) and reduces the immune depression observed in SARS-CoV-2 infection. This element has a synergistic action with several antiviral drugs used in infections with various viruses (e.g., HIV, hepatitis C, and others). Zn deficiency lowers general immunity and, in particular, increases the risk of respiratory infections (like COVID-19) [254]. This effect of Zn also occurs for various reasons in immunocompromised patients with COVID-19 or in older adults in whom this immunity is usually lower [255].

### 7.2. Pharmacokinetic Interactions

Some drugs frequently used to treat gastric and duodenal ulcers such as H_2_ receptor antagonists (cimetidine, ranitidine) or neutralizing antacid medications (e.g., Maalox or aluminium hydroxide plus Mg hydroxide) modify the pharmacokinetics of azithromycin [256]. The antacid medication administered before cimetidine treatment did not affect the time to maximum concentration (T_max_) or area under the curve of plasma concentrations from 0 to 48 h after administration (AUC_0–48_), but significantly reduced the maximum plasma concentration reached (C_max_) by 24% [256]. Some antacids contain Mg or Zn salts. The use of antacid medication with azithromycin should be done with caution. In patients with COVID-19 treated with azithromycin, it is necessary to replace the gastric antacid medication (including the one containing Mg and Zn) with other medications for the treatment of gastric and duodenal ulcers. Moxifloxacin is one of the antibacterial antibiotics administered (without therapeutic success) to some patients with COVID-19. Antacids containing Mg and Zn salts (but also those containing aluminum salts) as well as H_2_ histamine receptor blockers significantly reduce the bioavailability of this antibiotic [257]. Mg trisilicate is another Mg compound with antacid action. Simultaneous oral administration of dexamethasone and Mg trisilicate significantly reduces dexamethasone absorption. In this case, reduction in the amount of absorbed dexamethasone is likely due to the fact that part of the drug is fixed on the surface Mg trisilicate [258]. Based on the existing data, dexamethasone should not be administered orally with or within a short time of therapeutic administration of Mg compounds. Chloroquine acts as an ionophore for Zn. Experimental studies on a human ovarian cancer cell line (A2780) showed that Zn entry into cells was increased when this cation was administered with chloroquine. After this effect of chloroquine, Zn^2+^ ions accumulate especially in lysosomes [259].

### 7.3. Influence of Anti-COVID-19 Medication on Plasma or Tissue Mg^2+^ and Zn^2+^ Concentrations

Some drugs used in anti-COVID-19 therapy produce hydroelectrolyte imbalances [260]. It is difficult to determine the exact influence of different drugs used in anti-COVID-19 therapy on plasma concentrations of Zn and Mg (as well as other elements) for the following reasons. First, when patients are admitted to the hospital, the concentration of these biometals is determined only intermittently. Second, COVID-19 influences these concentrations. Third, Zn and Mg concentrations are rarely determined during hospitalization and at discharge. Fourth, nutritional differences between patients during hospitalization (related to their specific clinical condition and feeding difficulties) lead to changes in concentration that are difficult to dissociate from the influence of the eventuality of the anti-COVID-19 medication. Finally, other drugs administered to patients for chronic preexisting conditions may also influence Mg and Zn concentrations.

### 7.4. Influence of Mg and Zn on Adverse Effects of Anti-COVID-19 Medication

One of the most severe adverse effects of anti-COVID-19 medication is QT-interval prolongation. This effect has been reported in some cases of treatment with lopinavir/ritonavir, hydroxychloroquine, and azithromycin [261]. This prolongation is associated with an increased risk of sudden death [262,263]. Another adverse effect of these drugs is torsade de pointes. In some cases, torsade de pointes is followed by ventricular fibrillation and death. Both chloroquine and hydroxychloroquine block the KCNH2-encoded HERG/Kv11.1 potassium channel and can prolong the QT interval. Hypomagnesemia (Mg < 1.7 mg/dL) is a risk factor for both torsade de pointes and QT-interval prolongation [262]. Azithromycin also increases the risk of cardiac arrest and sudden death [264,265]. Torsade de pointes is associated with QTc-interval prolongation [265]. In these patients with COVID-19, the risk of sudden death is higher than in other individuals [262]. Hypomagnesemia aggravates this situation. In both situations, Mg deficiency is involved in the pathogenesis of this adverse effect. Mg administration is strictly necessary in these patients. Patients with congenital QTc prolongation syndrome have a particular situation [265]. There are many cases in which the patient with COVID-19 does not declare this fact. Administration of azithromycin, chloroquine, or hydroxychloroquine is not indicated for these patients. If the administration of these drugs has started, it must be stopped immediately and replaced with other drugs with anti-COVID-19 action. Lopinavir/ritonavir induces bradycardia in some patients. This effect has been observed both in patients with COVID-19 and in other patients such as those with HIV infection and treated with these drugs [266]. Existing statistical data confirm a highly significant association between age > 65 years (at which hypomagnesemia and hypozincemia are more common) and some chronic diseases (e.g., diabetes mellitus) in which there is, in most cases, Zn and Mg deficiency with an increased risk of death in patients with COVID-19 [267].

## 8. Discussion, Conclusions, and Future Directions

In summary, nutritional Mg appears to be important in both the prevention and treatment of COVID-19 disease for several reasons:Low overall dietary Mg status correlates strongly with several risk factors for COVID-19 [22,23,24,26,268,269,270].COVID-19 infection risk is shown to be higher in areas of low environmental Mg [105].There is a high prevalence of hypomagnesemia at hospital admission of patients with SARS-CoV-2 infection [271,272].Hypomagnesemia seems to be an independent risk factor associated with unfavorable prognosis in hospitalized patients with COVID-19 [268,271,272,273].Increased dietary intakes of Mg have often been shown to be protective in patients with COVID-19 [86,87].Pulmonary inhalation of Mg improves oxygenation in patients with COVID-19 [126].Low Mg status and hypomagnesemia are associated with the neuropsychiatric complications seen with COVID-19 (see Section 6 in this review).SARS-CoV-2 infection might be a cause of hypomagnesemia (this review).Mg as well as Zn may be beneficial in COVID-19 therapy by enhancing the effects of medications or diminishing their side effects (see Section 7 in this review).

Given these findings, it is not surprising that Mg has often been mentioned as a promising adjunct strategy in the prevention and treatment of COVID-19 [22,23,24,26]. However, despite these facts, no clinical trials of oral Mg supplements alone for COVID-19 have been reported.

Information gathered in this review suggests that the synergy of Mg with other essential nutrients may be important to its beneficial role, and comprehensive dietary Mg status studies have shown that high-Mg diets are protective in patients with COVID-19 [86,87]. The facilitation of enzymatic reactions—a primary role of nutrients—is often accomplished via the synergy of multiple nutrients [19], suggesting that a combination of essential nutrients given as an intervention may have different outcomes than single nutrient therapy [20]. Thus, Mg trials present promising areas for future research, both as a solo nutrient as well as with nutritional research designed to discern the impact of not only Mg but also its interaction with other micronutrients. Such studies are especially needed regarding the implications of the disruption of normal or optimal dietary and circulating ratios in patients with COVID-19 between Mg and other bivalent cations (e.g., zinc and copper), other micronutrients important in patients with COVID-19 (e.g., vitamin D), as well as electrolytes. Hypokalemia, hypocalcemia, hyponatremia, and hypernatremia have been noted in patients with COVID-19 [7,213,274,275,276], all of which have been long known to be related to hypomagnesemia [277]. Of twenty-two studies in PubMed on electrolytes in COVID, only four include mention of Mg, and only two of these show its impact [75,278]. A full study of electrolyte balance in COVID-19 that includes Mg is an important topic that has been neglected and deserves careful study with standardized values for definition of hypomagnesemia [279].

Since Mg deficiency is involved in the occurrence and aggravation of respiratory and/or neuropsychiatric complications of COVID-19, systematic determination of circulating Mg concentration and prompt correction of its imbalance with other circulating cations is necessary to prevent and/or improve these CNS, respiratory, and other complications.

Regarding COVID-19 disease treatment, interrelationships between anti-COVID-19 medications and two of the main bivalent cations in the human body (Mg^2+^ and Zn^2+^) can be useful in increasing drug therapy effectiveness while reducing some adverse effects of anti-COVID-19 drugs. More clinical studies regarding the therapeutic effect of combining Mg and Zn with medications used to treat COVID-19 are necessary. The efficiency of the different compounds of Mg and Zn with varying pharmacokinetic parameters combined with anti-COVID-19 medication must also be studied. The effects of the administration of these two cations after discharge from the hospital of patients who have been treated for COVID-19 is another factor that must be considered.

Finally, we all need to keep in mind that many patients are admitted to the hospital with Mg deficits prior to COVID-19 onset, and those leaving the hospital after a bout with COVID-19 may take away a Mg deficit unless proper testing is administered, and Mg therapy prescribed.

## Figures and Tables

**Figure 1 biology-12-00735-f001:**
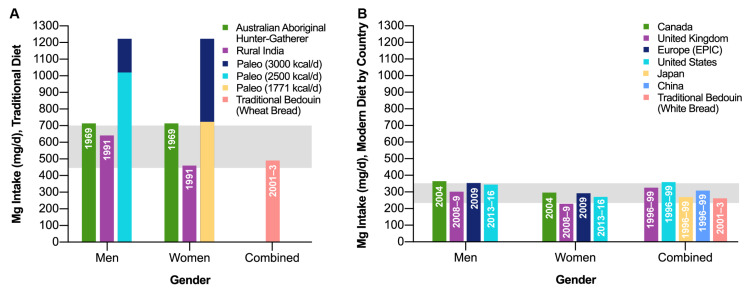
Comparison of mean dietary magnesium intakes among adults by diet type. (**A**) Traditional diet (450–700 mg Mg/day). (**B**) Modern diet (250–350 mg Mg/day). Shaded areas indicate range of mean adult Mg intakes of these various studies. Reprinted/adapted with permission from Rosanoff [83] for source data (figure adapted with author permission).

**Table 1 biology-12-00735-t001:** Low Versus Normal Serum Magnesium in COVID-19 Studies.

Mg Status	Serum Mg Level			COVID-19–Associated Outcomes
	mg/dL	mmol/L	mEq/L	
Severe hypomagnesemia	<1.58	<0.65	<1.3	Present in 13% of hospitalized patients with COVID-19 [52]
Hypomagnesemia	<2.0	<0.82	<1.65	Risk factor for severe COVID-19 [54]
	<1.82	<0.75	<1.5	Defines Mg deficiency; increased length of CCU stay [52]
	1.80	0.74	1.48	Infection severity and worsened prognosis in ICU patients with COVID-19 [55]
	≤1.8	<0.74	<1.48	Death [53]; predicts arterial thromboembolism [56]
High normal	2.19–2.26	0.90–0.93	1.8–1.86	Protective against deterioration [54]

Abbreviations: CCU, critical care unit; ICU, intensive care unit; Mg, magnesium.

**Table 2 biology-12-00735-t002:** Mechanisms of Action of Magnesium in Pulmonary Complications of COVID-19.

Pulmonary/Respiratory Complication of COVID-19	Possible Link with Mg
Respiratory system is the main organ system involved in COVID-19 [130]	Intravenous Mg sulfate has a clinical impact on acute severe asthma [129]
	Mg sulfate can dilate constricted pulmonary arteries and reduce pulmonary artery resistance as well as induce bronchodilation by inhibition of airway smooth muscle contraction [126]
	Mg sulfate extended infusion has been suggested as an adjunctive treatment for critically ill patients with COVID-19 [131]
V/Q mismatch in COVID-19 may be responsible for severe respiratory complications of COVID-19 [121,122,123]	
Inhalation therapies for COVID-19 suggested to improve oxygenation and V/Q mismatch [124,125]	Nebulized Mg sulfate therapy has been proposed to reduce V/Q mismatch and improve oxygenation [126]
	Nebulized Mg sulfate improved oxygenation in patients with severe COVID-19 with hypoxia with V/Q mismatch but less so for critically and very severely ill patients with COVID-19 with some degree of intrapulmonary shunt [126] [manuscript in preparation]

Abbreviations: Mg, magnesium; V/Q, ventilation–perfusion.

**Table 3 biology-12-00735-t003:** Magnesium in Neurological and Psychiatric Complications of COVID-19.

Neurological or Psychiatric Complication of COVID-19	Possible Link with Mg
**Memory and cognition** [147,148]	Mg plays a role in mechanisms of memory and cognition, and Mg deficit is involved in significant reduction in memory [149,150,159]
	Increasing brain Mg in rats is positively correlated with increased working, long-term, and short-term memory [151]
	Mg has been shown to be involved in prevention of memory loss or recovery of memory via increased neuroplasticity [151,156] upregulation of cAMP [23] modulation of transcription factors (c-Fos, NF-κB) [23,154] increased BDNF action [23] reduced oxidative stress in the brain [23] increased levels of reduced glutathione [23] reduced proinflammatory cytokine synthesis and release [23,155,161] reversed upregulation of TNF-α/NF-κB signaling in the hippocampus [154]
Impaired c-Fos activation [152]	Impaired activation of c-Fos is necessary for memory formation [152]
	Low Mg impairs c-Fos activity [153]
	Mg administration normalizes c-Fos expression [153]
Neuroplasticity and synaptic interconnection mediate memory [156]	Mg improves synaptic plasticity [151]
	Mg plus environmental enrichment synergistically improved recognition/spatial memory by reducing synaptic loss and restoring the NMDAR signaling pathway in AD mice [159]
	Chronically reducing Ca^2+^ flux enhancement synaptic plasticity [151]
CREB-mediated transcription involved in long-and short-term memory [152]	Mg upregulated CREB-mediated transcription [158]
**Loss of taste and smell** [138,143,170]	TAS2R7 taste receptor is activated by Mg [167]
	Patients with COVID-19 and low Mg status showed change and/or loss of taste and/or smell [168,170]
**Loss of appetite** [85,130,172]	Mg deficiency causes loss of appetite [171]
**Ataxia** [138]	Mg depletion is associated with decreased cerebellar functions, including ataxia [24,173,174,175,176,177]
	Mg administration contributed to improvement in cerebellar clinical symptoms [173,174,175]
	There are no clinical trials of Mg therapy for patients with patients with COVID-19 presenting with ataxia, but we consider Mg administration should be considered and studied
**Impaired consciousness** [132,138]	Impaired consciousness occurs in about 14% of patients with patients with COVID-19 [132,138]
	Both hypo- and hyper-magnesemia sometimes cause severe disturbances of consciousness [179,180]
**Mental fatigue and inattention** [181,182]	These clinical manifestations were reversible with Mg administration [183]
Demyelination, cranial nerve palsy, and axonal neuropathies [184]	In patients with MS, Mg has protective action at the level of myelin [193,194,195,196]. MS demyelination is associated with low plasma as well as low cerebral spinal fluid Mg [197]; these low blood Mg levels are associated with decreased concentrations of other metallic elements [198]
	Data on Mg and metal trace elements in patients with patients with COVID-19 are few, but we consider that partial protective mechanism of Mg in MS is the same mechanism by which Mg can reduce demyelination in COVID-19 complications
**Convulsions or generalized seizures** [138,143,188]	Hypomagnesemia is associated with the production of convulsions, lower convulsive threshold, and increases in brain glutamate concentrations [25,183]
**Headache** [130,138,143,199]	Headache complication of COVID-19 related to direct viral invasion as well as cytokine release syndrome [130,199] and hypoxia [130], both of which are associated with low Mg status [115,126]
**Dizziness** [137,138,143]	Mg administration improves symptoms and alleviates dizziness [203,204]

Abbreviations: AD, Alzheimer’s disease; BDNF, brain-derived neurotrophic factor; CREB, cAMP response element-binding protein; Mg, magnesium; MS, multiple sclerosis; NF-κB, nuclear factor-κB; NMDAR, *N*-methyl-d-aspartate receptor; TNF, tumor necrosis factor.

## Data Availability

Not applicable.

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
