# Peer review of "Importance of Magnesium Status in COVID-19"

_biology, 2023, doi:10.3390/biology12050735_

Round 1

Reviewer 1 Report

The manuscript is well-written, informative and organized. However it is lengthy thus it is required to have tables, figures and concise conclusion part to summarize the review findings. It is highly recommended thus to add a graphical abstract and a table (please see below in detail) and reformulate the abstract and conclusion part.

Simple Summary

Lines 27-28: rephrase “In most modern cultures, magnesium intake is low and has been for decades” with “In most modern cultures, magnesium intake has been low for decades”

Line 32: replace “plans of study” with “study plans”

Abstract

Line 40: replace “Essential element magnesium (Mg) plays substantial…” with “Magnesium (Mg) is an essential element and plays substantial…”

Please consider rewriting some parts of the abstract to better capture the key points of the review

1.Introduction

Line 72: “Mg itself is a substrate”

Line 74: replace “The role Mg plays in” with “Mg’s role in”

Line 84: replace “to lead to decreased” with “to decrease”

Lines 95-96: replace “explain increased risk” with “explain the increased risk”

Line 108: Please consider metabolomics-based methods to track low biovailability of micronutrients https://www.ncbi.nlm.nih.gov/pmc/articles/PMC8234252/

2. Low Serum Mg and COVID-19

Line 137: replace “in the scale” with “on the scale”

2.1. Low Serum Mg and Risk of Developing Severe COVID-19 in Hospitalized Patients

Line 151: replace “or need for diuretic” with “or the need for diuretic”

Line 167: replace “study in” with “study of”

Lines 177-178: replace “both very high and low serum Mg levels” with “high and low serum Mg levels”

3. Low Dietary Mg Intake and COVID-19

Line 215: replace “There is strong evidence showing” with “Strong evidence shows”

Line 225: replace “pulmonary chronic disease” with “chronic pulmonary disease”

Line 227: replace “to be predictive of” with “to predict”

Line 232: replace “increase risk of death” with “increase the risk of death”

Line 245: replace “general mineralization of water” with “general water mineralization”

4. Mg Supplements and COVID-19

Lines 275-276: rephrase “Accordingly, Mg supplements were suggested for the prevention and treatment of COVID-19” with “Accordingly, Mg supplements were suggested to prevent and treat COVID-19”

Line 315: correct “in order to”

5. Mechanisms of Action of Mg in Pulmonary Complications of COVID-19

Line 323: replace “areas of the lung” with “lung areas”

Line 331: replace “that has” with “with”

Line 335: replace “accumulation in the lungs” with “lung accumulation”

Line 344: replace “Iran” with “Iran’s”

Lines 338-348 please refer to the study results

Line 348-349: replace “very severely” with “severely”

Line 351: replace “in addition to” with “and”

6. Mg in Neurological and Psychiatric Complication of COVID-19

Line 361: “Some of the complications”

Line 369: erase “among others”

6.2. Memory and Cognition

Line 419: replace “modulation of activity” with “modulation of the activity”

Line 427: replace “modulate activity” with “modulate the activity”

Lines 448-449: rephrase “CREB-mediated transcription is involved both in the mechanism of long-term memory and in short-term memory” with “CREB-mediated transcription is involved in both the mechanism of long-term and short-term memory”

Line 468: correct “BDNF to be one of…”

6.5. Confusion, Delirum, and Consciousness Disturbances

Line 502: replace “not known” with “unknown”

6.9. Headache and Dizziness

Line 567: replace “is entry of” with “is the entry of”

Line 568: replace “and increased synthesis” with “and the increased synthesis”

6.10. Immunity

Line 607: replace “in regard to” with “regarding”

7. Interrelationships Between Mg, Zn, and Agents Used to Treat COVID-19

Line 639: correct “Many of the drugs”

Line 642: correct “may possibly be”

7.1. Pharmacodynamic Interactions

Line 678: replace “to potentiation” with “to the potentiation”

Line 680: replace “in terms of” with “regarding”

Line 699: replace “together with” with “and”

Line 708: replace “Administration of” with “The administration of”

Line 716: erase “also”

Line 736: replace “could be involved in increasing” with “could increase”

Line 739: correct “fixing on this receptor” and “can penetrate in the cell”

Line 743: replace “proinflammatory” with “pro-inflammatory”

Line 751: replace “facilitate fixation” with “facilitate the fixation”

Line 772: replace “stimulates activity” with “stimulates the activity”

Line 773: replace “has an influence on” with “influences”

7.2. Pharmacokinetic Interactions

Line 812: replace “the reduction of” with “reduced”

7.3. Influence of Anti–COVID-19 Medication on Plasma or Tissue Mg2+ and Zn2+ Concentrations

Line 829: replace “the influence the eventuality” with “the influence of the eventuality”

7.4. Influence of Mg and Zn on Adverse Effects of Anti–COVID-19 Medication

Lines 845-846: replace “necessary in” with “necessary for”

Line 849: replace “If administration” with “If the administration”

Line 850: correct “must be replaced”

Lines 858-860 this statement needs to be rephrased. Is this compared to patients with non-covid, healthy individulas or non-hospitalized covid patients?

Line 863 : Please rephrase to “ as promising adjunct strategy

Line 865: The authors mention the need for clinical trials to study the effect of Mg administration on COVID-19 severity. However, as they mention also, the synergy of nutrients is important for their beneficial role.

Please shortly discuss the study design to address this. Consider this reference https://nutrition.bmj.com/content/3/2/419

8. Discussions, Conclusions, and Future Directions

Please add references in the discussion part

Line 875: replace “regarding implications” with “regarding the implications”

Line 879: correct “in human the body”

Line 883: replace “used in combination with” with “combined with”

Please include a summarizing table with the references presented in text regarding sections 5 and 6.

The authors have written an extensive and comprehensive review on the role of Megnesium in COVID-19. To make it more easy to read please include a diagram or table with the points that need to be considered in terms of upcoming studies and the open questions that warrant more studies.

Reviewer 2 Report

Comments to authors:

In this review, the authors are emphasizing the tentative role of Mg in COVID-19 disease and treatment. It is an interesting theory that magnesium status has relevance for the outcome of COVID-19 and the hypothesis that Mg could be protective during the disease course. However, I think that the huge literature material presented must be interpreted and described in a clearer way:

E.g. in the abstract: The hypotheses must be more specific and clearly underbuilt. E.g. the hypothesis that Mg could be protective must be followed by exactly what it seems to be protective against and also followed by the specific evidence for this effect. 

It should say "could be" protective and not "are protective" - not enough evidence to claim "are" as well as have not demonstrated that a tentative Mg effect can be separated from a tentative Ca effect

The introduction contains scattered information about all possible effects of Mg and is really not introducing the subject. Some of it could be moved to the specialized Mg-sections further ahead. Sentences such as “The role of Mg in cell and tissue metabolism is multifactorial. Clear evidence highlights the role of Mg as a key signaling element and metabolite in cell physiology” do not say anything at all. 

The introduction should start with introducing COVID-19 and SARS-COV-2 on a molecular level to help the reader understand the connection between this viral infection and the regulation of electrolytes such as Mg and Ca (Viral binding to ACE2, over-activation of the renin-angiotensin system etc.) 

Line 859: The statement “it is reasonable to assume that the viral infection might be a cause of hypomagnesia” – How about other electrolytes, is hypomagnesia in COVID-19 an isolated event? or is it a general decrease in electrolyte status caused by fever and increased transpiration? This should be discussed throughout the paper to put Mg in relation to other elctrolytes. The authors are touching the subject in section 7, which is however only focused on treatment.

On the same note, both Ca and Mg levels have been observed to be low during essential hypertension (increased ACE activity) and Ca and Mg deficiency has been proposed to play a role in hypertension. In fact, hypocalcemia has been described in COVID-19 patients. The authors should include the role of Ca in their analysis, our put forward arguments for why Calcium should not be considered in association with magnesium.

In the studies described regarding Mg status -was the low Mg status associated with normal status of other regulatory divalent cations (Ca, Zn, Fe)? And how was the potassium status (K) in relation to the low Mg status? 

Abstract: Please move the sentence (and rephrase accordingly) stating this is a consortium review to the acknowledgement section (+ Line 53):

 “several members of the 46 MaGNet Global Magnesium Project (MaGNet) describe the impact of Mg status on COVID-19” 
